# Rapid Identification of Nontuberculous Mycobacterium Species from Respiratory Specimens Using Nucleotide MALDI-TOF MS

**DOI:** 10.3390/microorganisms11081975

**Published:** 2023-07-31

**Authors:** Lan Yao, Xuwei Gui, Xiaocui Wu, Jinghui Yang, Yong Fang, Qin Sun, Jin Gu, Wei Sha

**Affiliations:** Clinic and Research Center of Tuberculosis, Shanghai Key Laboratory of Tuberculosis, Shanghai Pulmonary Hospital, Tongji University School of Medicine, Shanghai 200433, China; spectium1981@126.com (L.Y.); 13611914313@139.com (X.G.);

**Keywords:** nucleotide MALDI-TOF MS, nontuberculous mycobacterium, species identification, clinical respiratory sample, diagnose

## Abstract

We performed a prospective study to evaluate the diagnostic accuracy of nucleotide matrix-assisted laser desorption ionization time-of-flight mass spectrometry (MALDI-TOF MS) in identifying nontuberculous mycobacterium (NTM) from clinical respiratory samples. A total of 175 eligible patients were prospectively enrolled, including 108 patients diagnosed with NTM pulmonary disease (NTM-PD) and 67 control patients with other diseases. All specimens were subjected to acid-fast staining, liquid culture combined with MPT64 antigen detection, and a nucleotide MALDI-TOF MS assay. NTM cultures were also subjected to the MeltPro Myco assay for species identification. Altogether, the sensitivity, specificity, positive predictive value (PPV), and negative predictive value (NPV) of nucleotide MALDI-TOF MS were 77.8% (95% CI: 68.6–85.0%), 92.5% (82.8–97.2%), 94.4% (86.8–97.9%), and 72.1% (61.2–81.0%), respectively; these results were not statistically different from the results of culture + MPT64 antigen testing (75.0% [65.6–82.6%], 95.5% [86.6–98.8%], 96.4% [89.2–99.1%], and 70.3% [59.7–79.2%], respectively). In the identification of NTM species, of the 84 nucleotide MALDI-TOF MS positive samples, 77 samples (91.7%) were identified at the species level. Using culture + MeltPro Myco assay as the reference standard, nucleotide MALDI-TOF MS correctly identified 77.8% (63/81) of NTM species. Our results demonstrated that the nucleotide MALDI-TOF MS assay was a rapid single-step method that provided the reliable detection of NTM and identification of NTM species. This new method had the same sensitivity and specificity as the culture + MPT64 antigen method, but was much more rapid.

## 1. Introduction

Human diseases caused by nontuberculous mycobacteria (NTM) have increased in prevalence worldwide during recent years [1]. For example, a study of NTM pulmonary disease in Canada reported 29.3 cases/100,000 persons for the period of 1998 to 2002 and 41.3 cases/100,000 persons for the period of 2006 to 2010 [2]. In mainland China, a comprehensive review and meta-analysis of qualified data estimated the prevalence of NTM infections prior to 2015 was 6.3% among suspected tuberculosis patients [3,4,5]. Although diseases caused by NTM are not as common as those from the *Mycobacterium tuberculosis* (MTB) complex, patients with NTM infections typically have a long course of disease and a high rate of drug resistance and treatment failure.

Identification of the species and subspecies responsible for NTM infections is important so that appropriate therapeutic regimens can be administered, and species identification has a clear impact on patient outcome and prognosis [6]. Clinical laboratories use multiple molecular methods for the identification of NTM species, such as nucleic acid hybridization assays, line probe assays, DNA sequencing, real-time PCR, and matrix-assisted laser desorption/ionization time-of-flight mass spectrometry (MALDI-TOF MS), and each method has advantages and limitations [7]. 

Clinical laboratories commonly use proteomic MALDI-TOF MS, a technique that matches the protein profile of an isolate to a reference dataset. This method can identify pathogens isolated from a single colony at the species level with high efficiency, high accuracy, and low cost, and is therefore commonly used to identify Gram-negative and Gram-positive bacteria, including mycobacteria [8]. Improvements in several specific procedures have increased the discriminatory power of proteomic MALDI-TOF MS. For example, a recent study used proteomic MALDI-TOF MS to distinguish pulmonary tuberculosis in sputum samples and achieved a sensitivity of 80% and a specificity of 89% [9]. Moreover, MALDI-TOF MS can be used to test multiple biological samples, including urine [10,11] and cerebrospinal fluid [12,13]. Nevertheless, proteomic MALDI-TOF MS cannot be used for the identification of NTM at present [8].

Recent studies have used nucleotide MALDI-TOF MS, a new application that is based on unique and characteristic nucleotide profiles, for the identification of mutations that promote cancer [14,15]. Another study reported promising results from the use of this new method for the direct analysis of sputum samples and identification of MTB and mutations related to resistance [16]. Compared with phenotypic measurements of drug resistance, the consistency from nucleotide MALDI-TOF MS was 91.3% for detecting mutations in related genes [16]. However, in terms of the species identification of NTM, data from real-world studies using nucleotide MALDI-TOF MS assays remain scarce. 

Considering the high sensitivity, specificity, and accuracy of the nucleotide MALDI-TOF MS assay, we evaluated the accuracy of this method in identifying NTM from clinical respiratory samples.

## 2. Materials and Methods

### 2.1. Study Population

Prospective enrollment of patients with suspected NTM pulmonary infections who were admitted to Shanghai Pulmonary Hospital affiliated with Tongji University was performed from November 2020 to October 2021. The inclusion criteria were: (1) age over 14 years; (2) HIV-negativity; (3) pulmonary or systemic symptoms and imaging findings (nodular or cavitary shadows with bronchiectasis accompanied by multiple small nodules) leading to suspicion of NTM pulmonary disease; and (4) NTM pulmonary disease that was not treated during the previous 60 days. The exclusion criteria were: (1) diagnosis with both NTM and MTB infections and (2) inability to expectorate sputum and refusal of a bronchoscopy examination.

The diagnostic criteria for NTM pulmonary disease were according to the American Thoracic Society (ATS) guidelines (2020 edition) [17] and the NTM guideline for China [18]. During hospitalization, every patient was provided with several sputum tubes for other tests, including additional culture and GeneXpert MTB/RIF, after providing the required sputum samples as a routine procedure to improve diagnostic accuracy. A total of 175 patients were included, where 108 patients met the reference diagnostic criteria for NTM pulmonary disease and 67 control patients were diagnosed with non-NTM pulmonary disease. 

### 2.2. Sample Collection

Patients included in the study provided sputum samples at any time within 7 days of hospital admission. Each sputum sample (at least 5 mL) was collected over the course of 16 h. Different sputum samples were subjected to acid-fast staining, BACTEC MGIT 960 culture, and nucleotide MALDI-TOF MS detection.

Patients who were unable to provide sputum samples underwent bronchoscopy to obtain samples of bronchoalveolar lavage fluid (BALF). For collection of a BALF sample, sterile saline was instilled into the site of the lung lesion (based on imaging results) through the bronchoscope, and fluid was recovered into a sterile tube. The obtained sputum and BALF samples were used for subsequent acid-fast staining, BACTEC MGIT 960 culture, and nucleotide MALDI-TOF MS detection. Positive cultures obtained from the MGIT 960 culture were subjected to further analysis. The culture suspension was tested using the MPT64 antigen detection kit, and the culture sediment was identified by multi-color melting curve analysis.

### 2.3. Acid-Fast Staining, MGIT 960 Culture, MPT64 Antigen Assay, and MeltPro Myco Assay

Each collected specimen was liquefied and centrifuged to obtain a precipitate. The sediment smear was fixed for anti-acid bacilli (AFB) staining, followed by microscopy examination [19].

The specimens were inoculated into MGIT tubes and incubated in the MGIT960 instrument (BACTEC MGIT 960 culture; BD Microbiology Systems, Franklin Lakes, NJ, USA). For a positive culture, the culture supernatant was tested using the MPT64 antigen detection kit (Genesis Corporation, Hangzhou, China), and the culture products were analyzed by multicolor melting curve analysis (MeltPro Myco assay, Zeesan Biotech, Xiamen, China) for species identification exactly according to the instructions. This method can identify 19 NTM species [20]. 

### 2.4. Nucleotide MALDI-TOF MS Procedures

The nucleotide MALDI-TOF MS procedure used the Conlight Myco kit, which was established and performed by Shanghai Conlight Medical Laboratory Co., Ltd. (Shanghai, China) as previously described in detail [16,21]. Briefly, this procedure had five steps and can be completed in about 3 days. The first step used the PCR. A PCR mixture (Appendix A; 1.3 μL of ddH_2_O, 0.5 μL of 10× PCR buffer with 20 mM MgCl_2_, 0.4 μL of 20 mM MgCl_2_, 0.1 μL of 25 mM dNTP mix, 0.5 μL of 1 μM primer mix, 0.2 μL of 5 U/μL PCR enzyme, and 5 μL of 5 ng/µL DNA template) was added into the wells of a 96-well plate, and PCR was then performed. A blank control (2 µL ddH_2_O), negative control (2 µL DNA extraction eluate), and positive control were performed at the same time. 

The second step used the shrimp alkaline phosphatase (SAP) assay. A 2 µL aliquot of the SAP mixture (1.53 μL of ddH_2_O, 0.17 μL of SAP buffer, and 0.30 μL of SAP enzyme) was added into each well of a PCR plate, and the total volume was adjusted to 7 µL. The sealed plate was centrifuged at 4000× *g* for 5 s and then incubated at 37 °C for 40 min and 85 °C for 5 min. 

The third step used PCR extension (Appendix A). A 2 µL aliquot of the iPLEX extension mix (0.62 µL of ddH_2_O, 0.2 µL of iPLEX buffer, 0.2 µL of iPLEX termination mix, 0.94 µL of extension primer mix, and 0.04 µL of iPLEX enzyme) was added into each well of a PCR plate, followed by the extension reaction. 

The fourth step used sample desalination. Clean resin on a 96/15 mg dimple plate was air-dried for at least 10 min. Then, 41 μL of water was added into each well, followed by the application of sealing film and centrifugation (3200× *g*, 5 min). Then, 15 mg of resin was added to the reaction products according to the manufacturer’s protocol (Agena Bioscience, San Diego, CA, USA). The sealing film was applied, and the contents were mixed by inversion and rotated for 15 min. Centrifugation was then performed at 3200× *g* for 5 min. 

The fifth step used MS. The PCR product was transferred to the SpectroCHIP according to the MassARRAY Nanodispenser operating instructions. Automatic analysis of the results was performed using MassARRAY analysis software (MassArrayTyper Version 4.1) (Appendix A).

### 2.5. Statistical Analysis

The values for sensitivity, specificity, positive predictive value (PPV), negative predictive value (NPV), and accuracy with 95% confidence intervals (CIs) were determined using SPSS version 19.0 (SPSS Inc., Chicago, IL, USA). Consistency was determined using the Kappa test, and the results were classified as almost perfect (Kappa ≥ 0.8), substantial (0.6 < Kappa < 0.8), moderate (0.4 < Kappa < 0.6), fair (0.2 < Kappa < 0.4), or slight (Kappa < 0.2). The χ^2^ test, Fisher’s test, or the *t*-test was used to verify the significance of differences between the NTM pulmonary disease group and the control group.

### 2.6. Ethical Approval

All procedures were approved by the Ethics Review Committee of Shanghai Pulmonary Hospital (L21-002). Written informed consent was obtained from all participants upon enrollment. This study was conducted according to the principles of the Declaration of Helsinki of the World Medical Association and according to Good Clinical Practice Guidelines.

## 3. Results

### 3.1. Study Design and Characteristics of Enrolled Cases

We evaluated the value of nucleotide MALDI-TOF MS technology in identifying NTM in patients by conducting a prospective study at Shanghai Pulmonary Hospital from November 2020 to October 2021 (Figure 1 and Figure 2). According to the inclusion and exclusion criteria, we enrolled 175 patients with suspected or confirmed NTM pulmonary disease after they provided informed consent. All diagnoses strictly followed the NTM pulmonary disease criteria established by the 2020 ATS clinical practice guideline [17] and the criteria for NTM pulmonary disease in China [18]. There were 108 patients in the NTM pulmonary disease group and 67 patients in non-NTM pulmonary disease (control) group. Each included patient provided sputum samples or a BALF sample, which was subjected to analysis by acid-fast staining, MGIT 960 culture, and nucleotide MALDI-TOF MS. For culture-positive samples, MPT64 antigen detection and multicolor melting curve analyses were also performed; the first test was for NTM detection, and the second test was for NTM species identification (Figure 1).

The NTM pulmonary disease group and the control group had no significant differences in age, gender, or major symptoms. However, the chest CT results showed a higher incidence of nodules (*p* = 0.010) and bronchiectasis (*p* = 0.021) in the NTM pulmonary disease group. Additionally, the percentage of sputum samples was greater in the control group (73.1% vs. 51.9%, *p* = 0.005) (Table 1).

### 3.2. Diagnostic Performance of NTM Identification by Nucleotide MALDI-TOF MS

Based on the reference standard of clinical diagnosis, the sensitivity, specificity, positive predictive value (PPV), negative predictive value (NPV), and accuracy of nucleotide MALDI-TOF MS for detection of NTM were 77.8% (95% CI: 68.6–85.0%), 92.5% (82.8–97.2%), 94.4% (86.8–97.9%), 72.1% (61.2–81.0%), and 83.4% (76.9–88.5%), respectively. The corresponding values for culture combined with MPT64 antigen detection were 75.0% (65.6–82.6%), 95.5% (86.6–98.8%), 96.4% (89.2–99.1%), 70.3% (59.7–79.2%), and 82.9% (76.3–88.0%), respectively. The corresponding values for acid-fast staining microscopy were 47.2% (37.6–57.0%), 80.6% (68.8–88.9%), 79.7% (67.4–88.3%), 48.7% (39.1–58.3%), and 60.0% (52.3–67.2%), respectively (Table 2). Thus, nucleotide MALDI-TOF MS detection of NTM performed similarly to culture combined with MPT64 antigen detection (no statistically significant differences), but both of these methods were significantly better than acid-fast staining microscopy. Using the Kappa test to assess the consistency of the three methods with the reference standard of clinical diagnosis, the Kappa for culture combined with MPT64 antigen detection was 0.66, and the Kappa for nucleotide MALDI-TOF MS was 0.67, indicating substantial consistency for both methods. However, the Kappa for acid-fast staining microscopy was only 0.25, indicating fair consistency (Table 2). 

We then analyzed the ability of nucleotide MALDI-TOF MS and culture combined with MPT64 antigen testing to detect NTM in patients with positive or negative acid-fast staining microscopy results. In patients with positive acid-fast staining microscopy results, the sensitivity of culture combined with MTP64 antigen detection was 92.2% (80.3–97.5%), the same as nucleotide MALDI-TOF MS. In patients with negative acid-fast staining microscopy results, the sensitivity of culture combined with MTP64 antigen detection was 59.7% (45.8–72.2%) and that of nucleotide MALDI-TOF MS was 64.9% (51.1–76.8%). There were no statistically significant differences in the sensitivity of the two detection methods in patients with either positive or negative acid-fast staining microscopy results (Table 3). The AUCs of the ROC curves for acid-fast staining microscopy, culture combined with MPT64 antigen detection, and nucleotide MALDI-TOF MS were 0.639 (0.556–0.722), 0.853 (0.794–0.911), and 0.852 (0.792–0.911), respectively (Table 2 and Figure 3). The results from nucleotide MALDI-TOF MS detection were consistently comparable to those from culture combined with MPT64 antigen detection, but the nucleotide MALDI-TOF MS technique required significantly less time.

### 3.3. Identification of NTM Species by Nucleotide MALDI-TOF MS

In theory, nucleotide MALDI-TOF MS can detect any NTM species based on the presence of specific single nucleotide polymorphisms (SNPs). We found that nucleotide MALDI-TOF MS was able to detect most clinically common and disease-associated NTM species. To evaluate its ability to identify NTM species, we performed on samples from the NTM pulmonary disease group. Of the 84 nucleotide MALDI-TOF MS positive samples, we identified 77 samples (91.7%) at the species level. The most common species were *M. abscessus* (36.9%, 31/84), *M. intracellulare* (35.7%, 30/84), and *M. kansasii* (9.5%, 8/84) (Figure 4a). At the same time, we also performed species identification on cultures using a melting curve analysis as the reference standard. Of the 81 culture-positive samples, we identified 79 samples (97.5%) at the species level (total of 8 species), and the most common species were *M. intracellulare* (40.7%, 33/81), *M. abscessus* complex (38.3%, 31/81), and *M. avium* (9.9%, 8/81) (Figure 4b). 

To assess the consistency of bacterial identification from nucleotide MALDI-TOF MS and melting curve analysis, we used 81 culture-positive NTM samples for an in-depth analysis. Among them, 18 samples (22.2%) yielded inconsistent results, including 13 samples (16.0%) that had positive culture results and negative nucleotide MALDI-TOF MS results; five samples (6.2%) had completely inconsistent results with two methods. Two samples were identified as *M. avium* by culture and melting curve analysis, but nucleotide MALDI-TOF MS detected *M. abscessus*. Three samples were identified as *M. intracellulare* by culture and melting curve analysis, but nucleotide MALDI-TOF MS only identified *Mycobacterium* sp. (Table 4). The consistency rate of the two methods was 77.8% (63/81).

## 4. Discussion

The clinical practice guidelines of numerous scientific societies recommend the use of validated molecular or mass spectrometry techniques for the identification of NTM isolates at the species level [17,22]. Nevertheless, proteomic MALDI-TOF MS can only be applied to pure cultures, and achieves better performance when using solid media instead of liquid media [23,24]. A recent study used proteomic MALDI-TOF MS of clinical samples to distinguish MTB from the non-TB but with a lower respiratory tract infection using sputum samples and reported a sensitivity of 80% and a specificity of 89% [9]. Nucleotide MALDI-TOF MS can also identify MTB from raw smear-negative cases, and its reported sensitivity was 61.9% and failure rate was 11.3% [16]. In our setting, nucleotide MALDI-TOF MS had a sensitivity of 77.8% and a specificity of 92.5% in detecting NTM, similar to the method of culture + MPT64 antigen testing. Our sensitivity was 64.9% for smear-negative NTM patients, similar to a prior report [16]. We also found that a single test missed 52.8% of NTM cases by smear and 20.4% of cases by culture. A previous study reported that missed culture-negative patients may progress to NTM lung infection within one year [25]. A positive smear result requires 5000 to 10,000 AFB/µL of sputum and a positive culture result requires 10 to 100 AFB/μL of sputum [26]. However, nucleotide MALDI-TOF MS has a detection limit of only about 10 MTB chromosomal DNA copies [16]. Two other advantages of nucleotide MALDI-TOF MS are that it has the same diagnostic performance as culturing and it is much faster than culture, in that the entire procedure only requires about 3 days.

It should be noted that our results regarding the distribution of different NTM species in China differ from previous studies. A 2013 nationwide survey of China reported the five most frequently isolated NTM species were the *M. abscessus* complex (36.0%), *M. avium-intracellulare* complex (34.1%), *M. kansasii* (9.8%), *M. paragordonae* (5.4%), and *M. lentiflavum* (3.2%) [4]. A 2016 systematic review showed that southern China had a significantly higher percentage of rapidly growing mycobacteria than northern China [3]. However, a 2020 study of 17 hospitals in China reported the dominant NTM species was *M. intracellulare*, followed by the *M. abscessus* complex [5]. In our study, the major NTM species was *M. intracellulare* (40.7%) followed by the *M. abscessus* complex (38.3%). Differences in the years of sample collection and the geographic locations of the study populations might account for these inconsistent results. Nevertheless, we confirmed that the *M. avium-intracellulare* complex and *M. abscessus* complex are still the predominant NTM in China.

Our results showed that nucleotide MALDI-TOF MS identified more than one species in 11 of the NTM samples and that two samples each had three NTM species. The MeltPro Myco assay only identified NTM-coinfection in 5 samples. We also identified MTB in three of the nucleotide MALDI-TOF MS samples (Figure 4a); two of these samples were BALF, and they all had a negative result in the GeneXpert MTB/RIF assay. Based on additional laboratory results, we ruled out a diagnosis of MTB in these three patients. False-positive results for MTB are common because dead MTB cannot be detected by culture and because China has a high prevalence of TB. Considering that nucleotide MALDI-TOF MS includes multiple PCR procedures and targets more than one target of MTB, the procedure should be used with caution in the diagnosis of active MTB, especially in patients with previous MTB infections and in patients from regions with a high TB burden. Our nucleotide MALDI-TOF MS results also identified other non-pathogenic species, *M. triviale* (*n* = 3) and *M. nonchromogenicum* (*n* = 1), which are only rarely responsible for disease in humans. NTM coinfection can lead to a more severe clinical outcome, and a previous study reported the incidence of coinfection was 13.5% [25]. Another study of patients receiving treatment for pulmonary *M. abscessus* infection reported that those with mixed infections had significantly worse outcomes [27]. It is hard to know if these coinfecting pathogens were clinically relevant or were just incidental and clinically irrelevant infections of the respiratory tract [28]. A repeated culture with an identification procedure is needed to determine if additional patients had coinfections that were undetected by culture [25]. 

Our study has several limitations. First, the MeltPro Myco test may underestimate NTM coinfection, and this will affect the consistency of the nucleotide MALDI-TOF MS in NTM species identification. Second, we did not perform a sequencing analysis of the five samples in which the two methods identified inconsistent NTM species, and this may have affected our reported consistency rate. Third, sputum cultures and nucleotide MALDI-TOF MS testing were not performed on the same samples, and this could have led to a lack of consistency. This study design required the collection of sputum samples within seven days of admission, and patients were carefully instructed on how to produce sputum samples, but there might have been some subtle differences among patients in the methods of sample collection [29].

## 5. Conclusions

In summary, nucleotide MALDI-TOF MS provides the rapid and sensitive detection of NTM from a single sputum sample or BALF sample, especially in culture-negative patients. Although nucleotide MALDI-TOF MS identified more species than the MeltPro Myco assay, there are several possible reasons for this difference, so these results should be interpreted with caution.

## Figures and Tables

**Figure 1 microorganisms-11-01975-f001:**
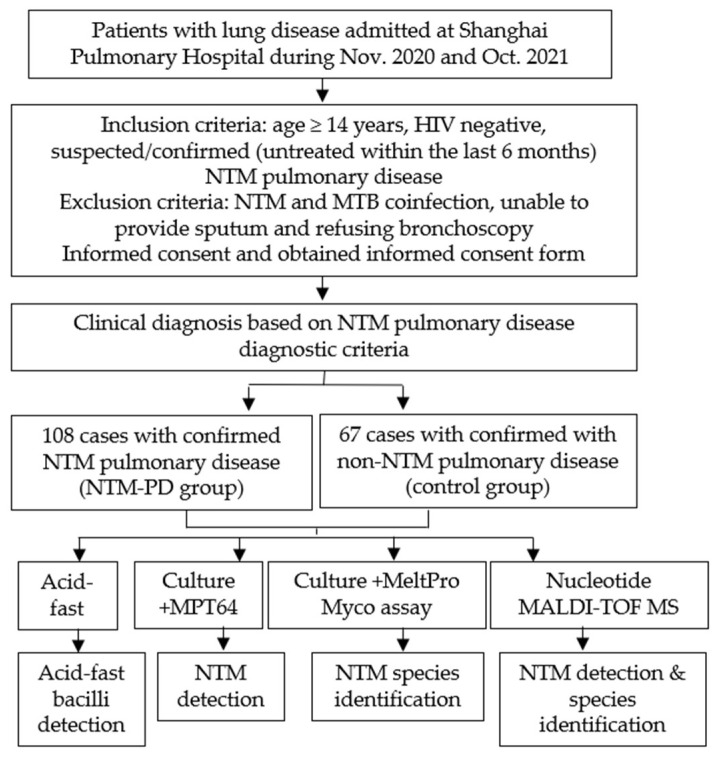
Study design.

**Figure 2 microorganisms-11-01975-f002:**
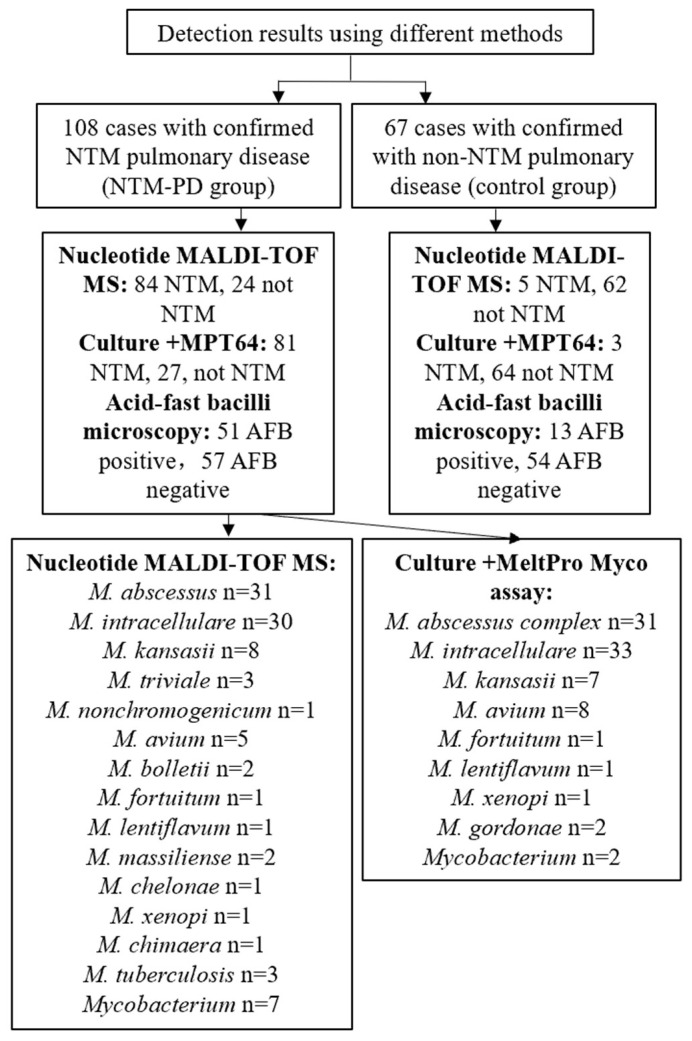
Results of NTM detection and species identification using different methods.

**Figure 3 microorganisms-11-01975-f003:**
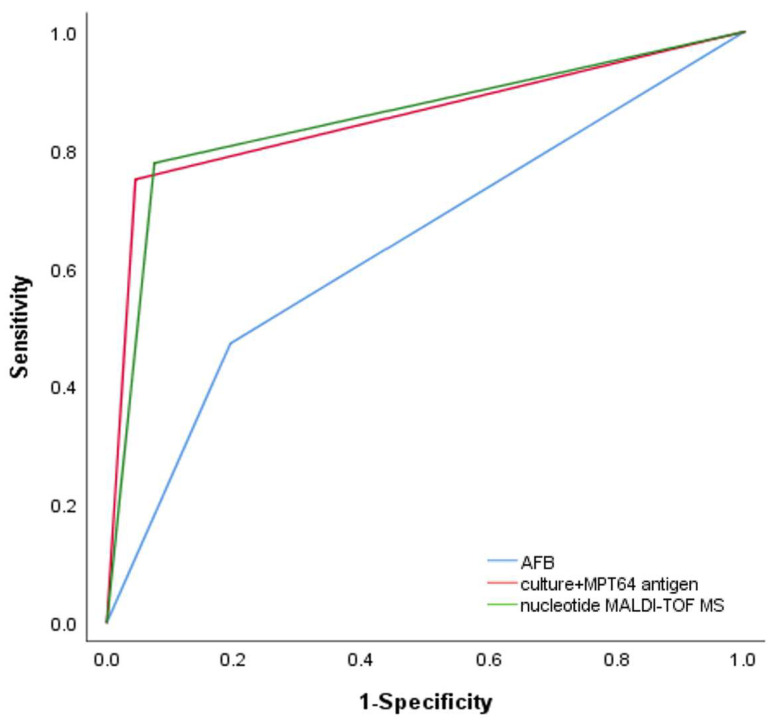
Receiver operating characteristic (ROC) curves for diagnosis of NTM using AFB (AUC: 0.639, 95% CI: 0.556, 0.722), culture + MPT64 antigen (AUC: 0.853, 95% CI: 0.794, 0.911), and nucleotide MALDI-TOF MS (AUC: 0.852, 95% CI: 0.792, 0.911).

**Figure 4 microorganisms-11-01975-f004:**
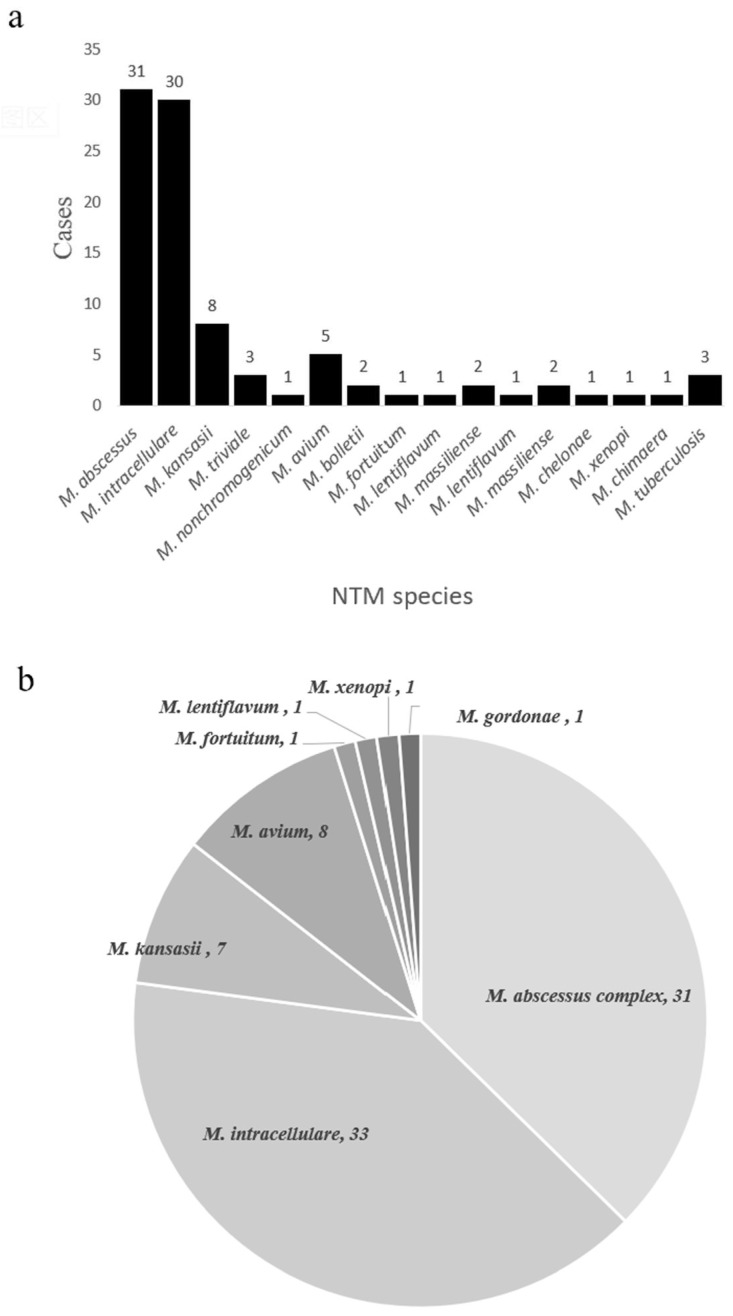
NTM species identified using the nucleotide MALDI-TOF MS assay (**a**) and the culture + MeltPro Myco assay (**b**).

**Table 1 microorganisms-11-01975-t001:** Demographic and clinical characteristics of the NTM pulmonary disease group and the control group.

	NTM Pulmonary Disease Group (*N* = 108)	Control Group (*N* = 67)	*p* Value
Age in years, median (range)	65 (24, 67)	59 (33, 68)	0.914
Male, *n* (%)	45 (41.7%)	30 (44.8%)	0.686
Major symptoms, *n* (%)			
Cough	85 (78.7%)	50 (74.6%)	0.523
Fever	11 (10.2%)	9 (13.4%)	0.512
Hemoptysis	23 (21.3%)	14 (20.9%)	0.950
Chest CT results, *n* (%)			
Nodule	93 (86.1%)	47 (70.1%)	0.010
Cavity	54 (50%)	33 (49.3%)	0.924
Bronchiectasis	85 (78.7%)	42 (62.7%)	0.021
Samples, *n* (%)			
Sputum	56 (51.9%)	49 (73.1%)	0.005

**Table 2 microorganisms-11-01975-t002:** Diagnostic performance of the AFB, culture + MTP64 antigen, and nucleotide MALDI-TOF MS methods.

Method	NTM Result	Clinical Diagnosis		Sensitivity, % (95% CI)	Specificity, % (95% CI)	PPV, % (95% CI)	NPV, % (95% CI)	Accuracy, % (95% CI)	Kappa (95% CI)	AUC (95% CI)
		NTM-PD (108)	Control (67)							
AFB microscopy	+	51	13	47.2 (37.6–57.0)	80.6 (68.8–88.9)	79.7 (67.4–88.3)	48.7 (39.1–58.3)	60.0 (52.3–67.2)	0.25 (0.12–0.37)	0.639 (0.556–0.722)
	−	57	54							
Culture + MPT64	+	81	3	75.0 (65.6–82.6) ^a^	95.5 (86.6–98.8) ^b^	96.4 (89.2–99.1)	70.3 (59.7–79.2)	82.9 (76.3–88.0)	0.66 (0.55–0.77)	0.853 (0.794–0.911)
	−	27	64							
Nucleotide MALDI-TOF-MS	+	84	5	77.8 (68.6–85.0) ^a^	92.5 (82.8–97.2) ^b^	94.4 (86.8–97.9)	72.1 (61.2–81.0)	83.4 (76.9–88.5)	0.67 (0.56–0.77)	0.852 (0.792–0.911)
	−	24	62							

NTM-PD: NTM pulmonary disease; PPV: positive predictive value; NPV: negative predictive value. “+”: NTM positive; “−”: NTM negative. ^a^ McNemar Test: *p* = 0.864 (Culture + MPT64 antigen vs. nucleotide MALDI-TOF MS); ^b^ McNemar Test: *p* = 0.727 (Culture + MPT64 antigen vs. nucleotide MALDI-TOF MS).

**Table 3 microorganisms-11-01975-t003:** Sensitivity of nucleotide MALDI-TOF MS and culture + MPT64 antigen in detecting NTM pulmonary disease in smear-positive and smear-negative patients.

Diagnostic Method	Smear-Positive	Smear-Negative
Culture + MPT64 antigen		
% (*n*/*n*)	92.2 (47/51)	59.7 (34/57)
95% CI	80.3, 97.5	45.8, 72.2
Nucl. MALDI-TOF MS		
% (*n*/*n*)	92.2 (47/51)	64.9 (37/57)
95% CI	80.3, 97.5	51.1, 76.8
*p* value	1	0.097

**Table 4 microorganisms-11-01975-t004:** Inconsistency of NTM species identification using different methods.

	Culture + MeltPro Myco Assay	Nucleotide MALDI-TOF MS
Inconsistent NTM species (*n* = 5, 6.2%)	*M. avium* × 2	*M. abscessus* × 2
	*M. intracellulare* × 3	*Mycobacterium* × 3
Culture + MeltPro Myco assay positive with Nucleotide MALDI-TOF MS negative (*n* = 13, 16.0%)	*M. intracellulare* × 4	negative
	*M. gordonae* × 2	
	*M. abscessus* complex × 2	
	*M. avium* × 2	
	*M. avium* + *M. intracellulare*	
	*M. lentiflavum*	
	*Mycobacterium*	

## Data Availability

The data that support the findings of this study are available from the corresponding author upon reasonable request.

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
