# Peer review of "Rapid Identification of Nontuberculous Mycobacterium Species from Respiratory Specimens Using Nucleotide MALDI-TOF MS"

_microorganisms, 2023, doi:10.3390/microorganisms11081975_

Round 1
Reviewer 1 Report
Lan Yao reported the use of MALD-TOF mass spectrometry in the rapid detection of Nontuberculous mycobacterium from the respiratory specimen in the current manuscript. Though it is not a newly invented approach, there are several reports available in which MALDI-TOF was utilized in the identification of Nontuberculous mycobacterium, nevertheless, this is an important region domain study. This manuscript can be accepted in the microorganism after addressing the following concern.
· Nontuberculous mycobacterium bacteria belong to diverse species and have similar protein composition, in this case, identification through MALDI-TOF is not a very adequate approach. It is recommended that authors include and discuss if they noticed any specificity in terms of detection.
Author Response
We thank the reviewer for raising this issue. The similar protein composition of nontuberculous mycobacteria makes it very challenging to use proteomic MALDI-TOF MS for identification. This is the reason we used the more recent technique of nucleotide MALDI-TOF MS for identification. As noted in our Introduction, previous studies have used nucleotide MALDI-TOF MS for identification of Mycobacterium tuberculosis. We used this technique because of its high sensitivity, specificity, and accuracy in identification of nontuberculous mycobacteria.Reviewer 2 Report
The "Nucleotide MALDI-TOF MS" diagnostic technique for the identification of non-tuberculous Mycobacteria directly on samples presented by Yao et al. seems interesting with a definite time saving, but the methodology of the study, although prospective, remains to be improved.
Major comments:
- the authors refer to reference 17 "treatment of Nontuberculous Mycobacterial Pulmonary disease" in the material and methods (l. 83) to define patients with confirmed NTM pulmonary disease. However, in the Result section (l. 155-157) they refer to the criteria of the American Society of Microbiology (2007), which I agree, are included in reference 17, but also to the diagnostic criteria in China, without specifying the reference. In the criteria of reference 17, the diagnosis of NTM disease must include clinical, radiological AND microbiological criteria, i.e. positive culture on > 1 sputum or 1 "deep" respiratory sample. In this case, I fail to see why, in the group of 108 patients with "confirmed NTM pulmonary diseases", only 86 had positive cultures. The others, who did not have positive cultures, by definition, should not be in the "confirmed NTM pulmonary disease" group. As a result, the sensitivity, specificity, PPV and NPV of nucleotide MALDI-TOF MS should be re-calculate.
- At the end (l. 303 – 304), we learn that the culture and MALDI TOF MS nucleotide were performed on different sputum samples! It's clear that this can have an impact on the results. This should be mentioned in the materials and methods.
- What about the 3 M. tuberculosis detected by MALDI TOF MS when the authors clearly excluded MTB+NTM co-infections (Figure 1, Figure 4a)? Are these false positives for MTB? The authors speculate in the discussion that these could be dead BK (l. 286 – 287). Did they have access to patient histories to verify this claim? Had dead BK had been detected by conventional PCR too?
- Regarding the discrepancies in NTM identification between Culture + MeltPro and nucleotide MALDI-TOF MS (Table 4), reference identification by gene amplification/sequencing (hsp65 or others) seems essential to decide who is right.
- Figure 4 b: in the pie chart showing the species identified by Culture + meltPro, there is the species Myc abscessus but previously in figure 2 it was noted Myc abscessus complex. If Culture + meltPro does not differentiate between the subspecies of Myc abscessus, please standardise to Myc abscessus complex. If this is the case, it would be interesting to check the subspecies of Myc abscessus found by MALDI TOF MS nucleotide by a gene amplification/sequencing technique. Idem for the samples identified Mycobacterium (Figure 1).
- Of the way the study is constructed, there are nucleotide MALDI TOF MS samples positive with a culture negative. These had to be listed in Table 4 as well and investigated (decapitation of the culture by pre-treatment? false positives of the MALDI-TOF nucleotide?).
Minor comments:
- Introduction: delete "and this coincides with a devreasing worldwide prevalence of TB" (l. 32 -33) because post-covid data tend to show the opposite and that the number of TB cases worldwide will increase.
- Introduction: l. 55 NTM-PD --> PD to be defined
- In the material and method section, specify the method for staining AFB (l. 100)
- In the discussion, it would be interesting to discuss the order of magnitude of the time saved by the nucleotide MALDI TOF MS technique versus culture.
Author Response
Reviewer #2 (Comments to the Author):
Comments to the Author
The "Nucleotide MALDI-TOF MS" diagnostic technique for the identification of non-tuberculous Mycobacteria directly on samples presented by Yao et al. seems interesting with a definite time saving, but the methodology of the study, although prospective, remains to be improved.
Major comments:
- the authors refer to reference 17 "treatment of Nontuberculous Mycobacterial Pulmonary disease" in the material and methods (l. 83) to define patients with confirmed NTM pulmonary disease. However, in the Result section (l. 155-157) they refer to the criteria of the American Society of Microbiology (2007), which I agree, are included in reference 17, but also to the diagnostic criteria in China, without specifying the reference. In the criteria of reference 17, the diagnosis of NTM disease must include clinical, radiological AND microbiological criteria, i.e. positive culture on > 1 sputum or 1 "deep" respiratory sample. In this case, I fail to see why, in the group of 108 patients with "confirmed NTM pulmonary diseases", only 86 had positive cultures. The others, who did not have positive cultures, by definition, should not be in the "confirmed NTM pulmonary disease" group. As a result, the sensitivity, specificity, PPV and NPV of nucleotide MALDI-TOF MS should be re-calculate.
Response: We thank the reviewer for this constructive comment. The microbiologic diagnostic criteria of NTM (positive culture results from at least two separate expectorated sputum samples or positive culture results from at least one bronchial wash or lavage) are the same in the 2007 and 2020 ATS guidelines and in China’s NTM guidelines. We added references to these in the Methods and Result sections of the revised version.
Regarding the specific query, there are two reasons for the presence of culture-negative patients in the "confirmed NTM pulmonary disease" group. First, we enrolled patients with confirmed NTM pulmonary disease who were not treated during the previous 60 days (Inclusion criterion No. 3). After review of the text, we noticed that our explanation of the second reason was not well written. We are therefore providing a better explanation here and in the Methods. To compare the results from nucleotide MALDI-TOF MS and culture, we unified the clinical sample types for both methods. The culture results refer to the same sample as was used in the nucleotide MALDI-TOF MS assay. We did not include other samples, which were also used for the clinical diagnosis of NTM-PD. Thus, patients with confirmed NTM pulmonary disease met all the diagnostic criteria. In this study, our purpose was to check the diagnostic performance of different methods using a single clinical sample.
- At the end (l. 303 – 304), we learn that the culture and MALDI TOF MS nucleotide were performed on different sputum samples! It's clear that this can have an impact on the results. This should be mentioned in the materials and methods.
Response: We thank the reviewer for this constructive comment. For some sputum samples, there were slight inconsistencies between the results from culture and the nucleotide MALDI-TOF MS assay. In some cases, we used different sputum samples from a patient for culture, acid-fast staining, and nucleotide MALDI-TOF MS. To minimize the differences between samples, patients were asked to collect each sputum sample over the course of 16 h. We revised the Methods section to clarify this issue.
- What about the 3 M. tuberculosis detected by MALDI TOF MS when the authors clearly excluded MTB+NTM co-infections (Figure 1, Figure 4a)? Are these false positives for MTB? The authors speculate in the discussion that these could be dead BK (l. 286 – 287). Did they have access to patient histories to verify this claim? Had dead BK had been detected by conventional PCR too?
Response: We thank the reviewer for mentioning this detail. Of the three M. tuberculosis samples, two were from bronchoalveolar lavage and the other was from sputum. We performed GeneXpert MTB/RIF on different sputum samples of each patient and they were all negative, although one had a positive T-SPOT result. Because some patients were uncertain about whether they had a previous MTB infection, there is a possibility of a previous missed screening. Considering that nucleotide MALDI-TOF MS includes multiple PCR reactions, targets more than one target of MTB, and uses other laboratory results, we ruled out a diagnosis of MTB in each of these three patients. As mentioned in the revised Discussion section, caution should be used in the diagnosis of active MTB in patients with previous MTB infections or patients from regions with a high TB burden.
- Regarding the discrepancies in NTM identification between Culture + MeltPro and nucleotide MALDI-TOF MS (Table 4), reference identification by gene amplification/sequencing (hsp65 or others) seems essential to decide who is right.
Response: We thank the reviewer for this comment. We agree with the reviewer that it is essential to use sequencing to determine which method is correct. However, we did not perform sequencing analysis of the five samples in which NTM species identification were inconsistent from the two methods. We mentioned this as a limitation of our study in the Discussion section. We plan to address this issue in future studies with larger sample sizes.
- Figure 4 b: in the pie chart showing the species identified by Culture + meltPro, there is the species Myc abscessus but previously in figure 2 it was noted Myc abscessus complex. If Culture + meltPro does not differentiate between the subspecies of Myc abscessus, please standardise to Myc abscessus complex. If this is the case, it would be interesting to check the subspecies of Myc abscessus found by MALDI TOF MS nucleotide by a gene amplification/sequencing technique. Idem for the samples identified Mycobacterium (Figure 1).
Response: We thank the reviewer for mentioning this detail. We apologize for this mistake. The species in Figure 4b should be M. abscessus complex, as indicated in the revised manuscript. We agree it would be meaningful to check the subspecies of M. abscessus found by nucleotide MALDI TOF MS by gene amplification and sequencing in a future study. So as the unidentified Mycobacterium species in Figure 1
- Of the way the study is constructed, there are nucleotide MALDI TOF MS samples positive with a culture negative. These had to be listed in Table 4 as well and investigated (decapitation of the culture by pre-treatment? false positives of the MALDI-TOF nucleotide?).
Response: We thank the reviewer for this comment. In our study, as indicated in Table 4, we used 81 culture-positive NTM cases for in-depth analysis. Thus, we did not list the cases with positive nucleotide MALDI-TOF MS results and negative culture results. But it would be interesting to look into those samples.
Minor comments:
- Introduction: delete "and this coincides with a devreasing worldwide prevalence of TB" (l. 32 -33) because post-covid data tend to show the opposite and that the number of TB cases worldwide will increase.
Response: We thank the reviewer for this constructive comment. We deleted this sentence in the revised manuscript.
- Introduction: l. 55 NTM-PD --> PD to be defined.
Response: We thank the reviewer for this attention to detail. We defined PD in the revised manuscript.
- In the material and method section, specify the method for staining AFB (l. 100).
Response: We thank the reviewer for this constructive comment. We added a reference in the revised manuscript.
- In the discussion, it would be interesting to discuss the order of magnitude of the time saved by the nucleotide MALDI TOF MS technique versus culture.
Response: We thank the reviewer for the constructive advice. We added this information in the revised Methods and Discussion sections.
Round 2
Reviewer 2 Report
- The authors' responses are satisfactory